# Macro-Micro-Coupling Simulation and Space Experiment Study on Zone Melting Process of Bismuth Telluride-Based Crystal Materials

**Huxiang Xia [1], Xiaoya Li [2],* and Qingyan Xu [1],***

[1] Key Laboratory for Advanced Materials Processing Technology (MOE),
School of Materials Science and Engineering, Tsinghua University, Beijing 100084, China;
xiahx18@mails.tsinghua.edu.cn
[2] Shanghai Institute of Ceramics, Chinese Academy of Sciences, Shanghai 201899, China
* Correspondence: xyli@mail.sic.ac.cn (X.L.); scjxqy@mail.tsinghua.edu.cn (Q.X.)

**Abstract:** Zone melting is one of the main techniques for preparing bismuth telluride-based crystal thermoelectric materials. In this research, a macro-micro-coupled simulation model was established to analyze the distribution of temperature and heat flow during the zone melting process. The simulation results show the melting temperature tends to affect the length of the melting zone, while the moving velocity of the melting furnace tends to affect the curvature of the melting and solidification interface. There are two small plateaus observed in the temperature curve of the central axis of bismuth telluride ingot when the moving velocity of the heat source is higher than 20 mm/h. As the moving velocity of the heat source increases, the platform effect is becoming more obvious. Based on the simulation results, the zone melt experiments were carried out both under microgravity condition on the Tiangong II space laboratory and conventional gravity condition on the ground. The experimental results indicate that the bismuth telluride-based crystal prepared in microgravity tends to possess more uniform composition. This uniform composition will lead to more uniform thermoelectric performance for telluride-based crystals. In the space condition, the influence of surface tension is much higher than that of gravity. The bismuth telluride ingot is very vulnerable to the influence of surface tension on the surface morphology during the solidification process. If the solidification process is not well controlled, it will be easier to produce uneven surface morphology.

**Keywords:** bismuth telluride-based crystal material; zone melting; macro-micro-coupling simulation

## 1. Introduction

With the increasing attention to energy conservation and environmental protection worldwide, thermoelectric materials have attracted extensive attention from society. Thermoelectric materials can directly convert environmental heat differences into electric energy through the Seebeck effect, without any pollution discharge. Such excellent characteristics make them have great application prospects in many fields such as refrigeration and thermoelectric power generation [1–3]. The thermoelectric conversion performance of thermoelectric materials is generally evaluated by the dimensionless figure of merit $ZT = S^2 \sigma T / (\kappa_e + \kappa_l)$, where $S$, $\sigma$, $T$, $\kappa_e$, $\kappa_l$ represent the Seebeck coefficient, electrical conductivity, absolute temperature, electronic thermal conductivity, lattice thermal conductivity, respectively. In order to improve the thermoelectric figure of merit ZT, researchers around the world have carried out a lot of research work on the defects of thermoelectric materials [4–7], carrier concentration [8,9], and electronic band structure [10–13], which has promoted the rapid development of thermoelectric semiconductor materials.

The highest thermoelectric figure of merit ZT around room temperature is about 1 for a bismuth telluride-based compound, which makes it a mature and widely used thermoelectric semiconductor material [14]. The crystal structure of bismuth telluride

material belongs to the orthorhombic crystal system and presents a hexahedral layered structure along the c-axis direction. The atoms in the same layer are of the same type, and the atoms in the different layers are bound by covalent bonds and ionic bonds. The layered crystal structure of bismuth telluride leads to its anisotropy in properties such as electrical conductivity and thermal conductivity, which eventually result in the maximum value of the thermoelectric performance of bismuth telluride material along with the atomic level [15].

The zone melting technique is a common method for preparing crystal bismuth telluride materials. The bismuth telluride-based raw material is first melted and then directionally solidified with the directional moving melting furnace. After directional solidification, the crystal orientation of bismuth telluride-based material becomes consistent, and the element distribution is more uniform. However, in the preparation process, due to the poor control of temperature distribution and excessive local thermal stress during the solidification, the neat arrangement of atomic layers in the crystal structure easily causes the material to cleavage between atomic layers, leading to cracks and even fracture, and eventually results in economic losses. When the temperature gradient of the solidification interface is not controlled properly, the solidification interface curvature is vulnerable to be unstable, which leads to partial undercooling and nucleation at the wall end, and finally introduces a crack defect. Therefore, preparing a high-performance bismuth telluride-based crystal bulk material without defects has always been a focus in the thermoelectric semiconductor fabrication field [16,17].

The zone melting technique is the mainstream commercial method for preparing bismuth telluride-based crystal. The control of the solidification process is one of the key factors to obtain high-performance crystal bismuth telluride materials without defect. König et al. compared bismuth telluride-based crystal materials prepared by the zone melting method under conventional gravity on the ground and microgravity on the space station [18]. The results show that under microgravity condition, the unsteady flow of melt decreases, and the solidification is relatively stable. Stable solidification can effectively avoid the streaks defect caused by melt convection. However, there are too few reports about the zone melting technique of crystal bismuth telluride materials in microgravity to support a deeper conclusion, which needs to be further explored urgently.

At present, using various experimental methods, it is still difficult to predict the solidification interface morphology, temperature and heat flow changes, microstructure growth, and other information potentially affecting the thermoelectric properties of the bismuth telluride-based crystal material prepared by the directional solidification method. In fact, however, the numerical simulation method of directional solidification is relatively mature in many fields such as crystal preparation, which can accurately simulate the temperature and microstructure changes in the solidification process and thus greatly accelerate the research progress [19–22]. For temperature field simulation, the finite element method, including energy equation and flow equation, is generally adopted. Relevant commercial simulation software has been widely applied. Phase-field simulation method based on Ginzburg–Landau second-order phase transition theory and cellular automata simulation method have also been widely accepted in microstructure simulation. Chen et al. simulated the growth and directional solidification process of columnar bismuth telluride by using the cellular automata finite element method. The research proved that the shape of the solidification interface has an important influence on the growth direction and crystal grain size [23].

At present, a lot of work in the field of bismuth telluride thermoelectric materials focuses on regulating the thermoelectric figure of merit by changing the element content. However, there are few studies on the preparation process of bismuth telluride crystal materials with the simulation method, though it has a very important impact on the large-scale preparation of bismuth telluride-based crystal materials for reducing material costs and accelerating material application. In this study, a macro-micro-coupled simulation model was established for simulating the zone melting process of bismuth telluride-based crystal

materials. The heat flow distribution, solidification interface curvature, and microstructure growth during the zone melting process were simulated to study the effects of the melting temperature and melting furnace moving velocity on solidification. Based on the simulation results, the zone melt experiments were carried out both under microgravity condition on the Tiangong II space laboratory and conventional gravity condition on the ground. The composition distribution, surface quality, and Seebeck coefficient distribution of crystals fabricated under space microgravity and ground conventional gravity were studied.

## 2. Experiment

High-purity of Bi (5N), Sb (5N) and Te (5N) granules were weighed according to the stoichiometric ratio of $Bi_{0.52}Sb_{1.48}Te_3$. A special ampoule was customized to contain molten bismuth telluride material as shown in Figure 1. The space experimental furnace is custom-developed by Shanghai Institute of Silicate. There are heating units, pulling device and temperature detection system in the experimental furnace to maintain the directional temperature gradient. The zone melting experiments were carried out under conventional gravity condition on the ground and microgravity condition on the Tiangong II space laboratory with same experimental furnaces. The length of the solidified sample is 55 mm, and the diameter is 7.8 mm. The samples were characterized by a YXLON MU2000-D225 CT analysis system. The X-rays produced by the CT analysis system can penetrate ampoules and bismuth telluride materials, and provide the interior morphology of the ingots. The Seebeck coefficient was detected by the potential Seebeck microprobe developed by the German Panco company and the German Aerospace Center. The detection area is the axial section of ingot. The starting position of measurement is 5 mm from the top of the ingot. The axial scanning spacing is 1 mm, the radial scanning spacing is 0.5 mm, and the test temperature is 25 °C.

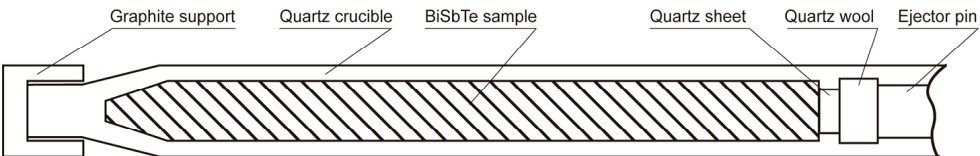

**Figure 1.** Customized ampoule structure.

## 3. Simulation

### 3.1. Macro Temperature Field Simulation Model

The heat exchange between the melting furnace and bismuth telluride based crystal material is one of the most important physical processes in the zone melting process. The heat exchange processes can directly affect the defect degree and thermoelectric properties of the bismuth telluride-based crystal materials. The zone melting process involves several specific heat exchange processes, such as thermal radiation from the melting furnace to the quartz ampoule, thermal conduction from the quartz ampoule to the material, and thermal convection between the surface of the quartz ampoule and the air. Three different heat transfer processes make the overall temperature solution more complicated. For solving heat exchange in the zone melting process, the Law of the Conservation of Energy is applied to describe the energy change of the system [24]:

$$\rho c_p \frac{\partial T}{\partial t} = \nabla \cdot (\lambda \nabla T) + \rho \Delta H \frac{\partial f}{\partial t} + Q_r \tag{1}$$

where $\rho$ is the material density, $c_p$ is the specific heat capacity, $T$ is the thermodynamic temperature, $t$ is the time, $\lambda$ is the thermal conductivity, $\Delta H$ is the latent heat of crystallization, and $f$ is the solid phase ratio of the grid during solidification. $Q_r$ is the heat flux density

between the current grid and reference grids. In different heat exchange processes, $Q_r$ has different forms. In thermal convection heat transfer, it can be expressed as:

$$Q_r = h\left(T - T_{ref}\right) \tag{2}$$

where $h$ is the heat conduction coefficient, $T$ and $T_{ref}$ are the temperature of the current grid and the temperature of the reference grids, respectively. In the process of radiation heat transfer, the Stefan–Boltzmann law is used to solve $Q_r$:

$$Q_r = \sigma\, T^4 \tag{3}$$

where, $\sigma$ is Stefan–Boltzmann constant, and the value is $5.67 \times 10^{-8}$W·m$^2$·K$^{-4}$.

The flow of liquid phase in the system can be guaranteed by the law of conservation of mass and the law of conservation of momentum (N-S equation):

$$\frac{\partial \rho}{\partial t} + \nabla \cdot \left(\rho \vec{v}\right) = 0 \tag{4}$$

$$\frac{\partial}{\partial t}\left(\rho \vec{v}\right) + \nabla \cdot \left(\rho \vec{v}\vec{v}\right) = -\nabla p + \rho \vec{g} + \vec{F}_s \tag{5}$$

where, $\vec{v}$ is the velocity, $p$ is the pressure, $\rho \vec{g}$ is the gravity term, and $\vec{F}_s$ represents the external force term.

### 3.2. Microstructure Growth Phase-Field Simulation Model

During the zone melting process, due to the high-temperature environment, it is very hard to directly observe the microstructure of bismuth telluride material. There are also few literature reports on the in situ observation experiment of bismuth telluride material. Numerical simulation is a more effective method for us to understand the evolution of microstructure. Phase-field method is one of the mainstream microstructure simulation methods based on Ginzburg–Landau second-order phase transition theory. After years of development, the phase-field method has been widely recognized by the academic community [25–28]. The free energy functional in the phase-field model can be expressed as the integral of calculating the free energy density function in the domain:

$$\mathrm{F}\left(\{\phi_\alpha\}, \left\{\vec{c}_\alpha\right\}\right) = \int_\Omega f\left(\{\phi_\alpha\}, \left\{\vec{c}_\alpha\right\}\right) \tag{6}$$

where $\phi_\alpha$ represents the content of solid phase, $\vec{c}_\alpha$ represents the corresponding element content. In the calculation, the free energy density function can be regarded as the combination of interface free energy density function and chemical free energy density function:

$$f\left(\{\phi_\alpha\}, \left\{\vec{c}_\alpha\right\}\right) = f^{intf}(\{\phi_\alpha\}) + f^{chem}\left(\{\phi_\alpha\}, \left\{\vec{c}_\alpha\right\}\right) \tag{7}$$

$$f^{intf}(\{\phi_\alpha\}) = \sum_{\alpha=1}^{m}\sum_{\beta=\alpha+1}^{m} \frac{4\sigma_{\alpha\beta}}{n\eta_{\alpha\beta}}\left(-\frac{\eta_{\alpha\beta}^2}{\pi^2}\nabla\phi_\alpha\nabla\phi_\beta + \phi_\alpha\phi_\beta\right) \tag{8}$$

$$f^{chem}\left(\{\varphi_\alpha\}, \left\{\vec{c}_\alpha\right\}\right) = \sum_{\alpha=1}^{n} \phi_\alpha f_\alpha\left(\vec{c}_\alpha\right) \tag{9}$$

where $m$ represents the number of phases involved, $\sigma_{\alpha\beta}$ and $\eta_{\alpha\beta}$ represents the interface energy and interface width between phase $\alpha$ and phase $\beta$. The variation of the phase-field variable with time $t$ can be obtained through variation of the free energy functional:

$$\frac{\partial \phi_\alpha}{\partial t} = -\sum_{\beta \neq \alpha}^{n} \frac{M_{\alpha\beta}}{m}\left(\frac{\delta F}{\delta \phi_\alpha} - \frac{\delta F}{\delta \phi_\beta}\right) \tag{10}$$

where $M_{\alpha\beta}$ is the interface mobility. The relationship between solute and time can be calculated by the quasi-equilibrium method [29,30]:

$$\frac{\partial c^i}{\partial t} = \nabla \left[ \sum_{\alpha}^{m} D_{\alpha}^i \varphi_{\alpha} \nabla c_{\alpha}^i + \sum_{\alpha \neq l}^{m} \frac{\eta_{\alpha l}}{\pi} \left( c_l^i - c_{\alpha}^i \right) \sqrt{\varphi_l \varphi_{\alpha}} \frac{\partial \varphi_{\alpha}}{\partial t} \frac{\nabla \varphi_{\alpha}}{|\nabla \varphi_{\alpha}|} \right] \tag{11}$$

where $D_{\alpha}^i$ is the diffusion coefficient of solute $i$ in phase $\alpha$.

### 3.3. Simulation Setup

The mature commercial software ProCAST was used to simulate the macroscopic temperature field in the zone melting process of bismuth telluride-based crystal materials. Based on the results of temperature field simulation, the temperature variation with time in the specified micro area is derived as the boundary condition of the microstructure simulation. The phase-field model coupled with the lattice Boltzmann method was used to simulate microstructure growth [28,31]. The GPU acceleration algorithm was applied to accelerate the simulation program. The size of the microstructure growth simulation is 1 mm × 1 mm, and the unit length of the mesh is 1 μm. The specific thermophysical parameters and boundary conditions are shown in Table 1.

**Table 1.** Thermophysical parameters and boundary conditions [23,32–35].

| Parameters | Unit | Values |
| --- | --- | --- |
| Ingot casting diameter | mm | 8 |
| Thermal conductivity of quartz | W/(m·K) | 7.1 |
| Thermal conductivity of bismuth telluride-based material | W/(m·K) | 1.2 |
| Density of quartz | g/m$^3$ | 2.65 |
| Density of bismuth telluride based-material | g/m$^3$ | 7.6 |
| Specific heat capacity of quartz | J/(g·K) | 1.1 |
| Specific heat capacity of bismuth telluride-based material | J/(g·K) | 0.19 |
| Heat transfer coefficient between quartz and bismuth telluride-based material | W/(m$^2$·K) | 500 |
| Latent heat of melting of bismuth telluride-based material | J/g | 232.9 |
| Melting point of bismuth telluride-based material | °C | 614.5 |
| Length of heating furnace | mm | 25 |
| Melting temperature | °C | 660 |
| Moving velocity of melting furnace | mm/h | 5 |
| Phased-field Interface width | μm | 4 |
| Solid–liquid Interfacial energy | J/m$^2$ | 0.132 |
| Solid–liquid Interfacial mobility | m$^4$/(j·s) | $4.0 \times 10^{-11}$ |

Zone melting of bismuth telluride-based material is essentially a process of component homogenization and crystallization of the material. In the zone melting process, a directional moving furnace provides continuous heat for the melting material. The heat is first transferred to the quartz crucible by means of thermal radiation and convection, and then to bismuth telluride material through heat conduction. Based on the actual zone melting process of bismuth telluride-based crystal material, the 3D heat transfer simulation model was established, as shown in Figure 2. The red part in Figure 2a shows that the solid phase fraction of the material is 0, which means that this region is all liquid phase. In order to verify the reliability of the model, a temperature measurement experiment was carried out. The experimental result and simulation result are shown in Figure 2b. The simulation results are in good agreement with the experimental results, indicating that the simulation model can accurately reveal actual temperature results.

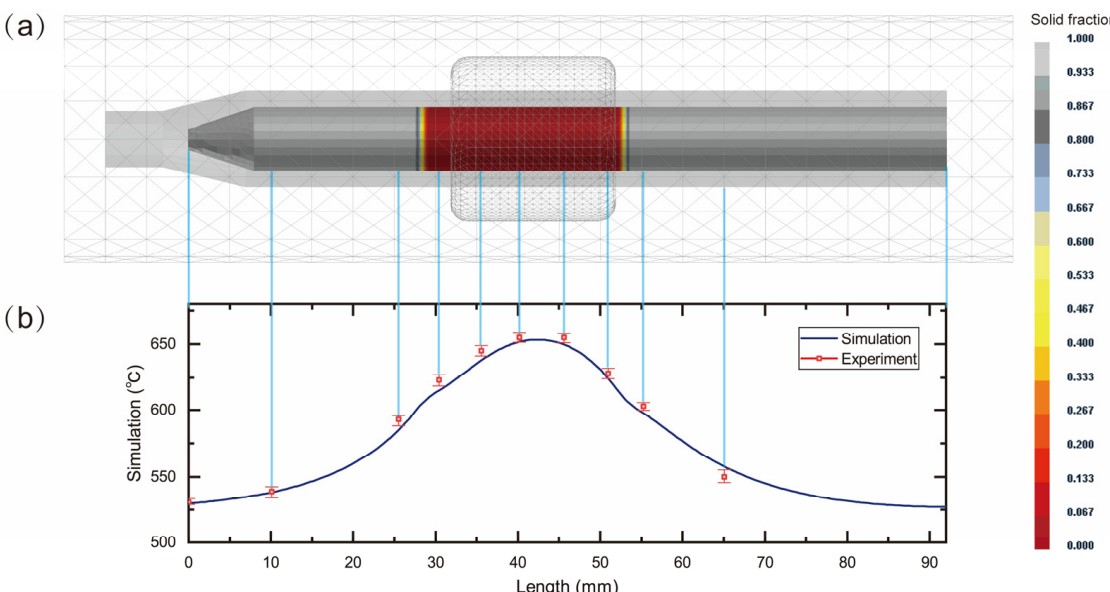

**Figure 2.** Temperature field modeling and experimental verification. (**a**) Solid phase fraction simulation during the zone melting process, the red part is the melting zone, (**b**) simulated temperature distribution and experimental temperature distribution.

The simulation results of the temperature field are introduced into the phase-field model for simulating microstructure growth. The microstructure growth simulation result is shown in Figure 3. Before the simulation, a layer of crystal nucleus is arranged at the bottom of the calculation domain, and then, the crystal structure will grow on the nucleus. At the beginning of the growth, the crystal extends upward in the form of cellular crystals. Then, the gap between cellular crystals is filled through liquid phase solidification. At about 3825 s, the crystal reaches a stable growth.

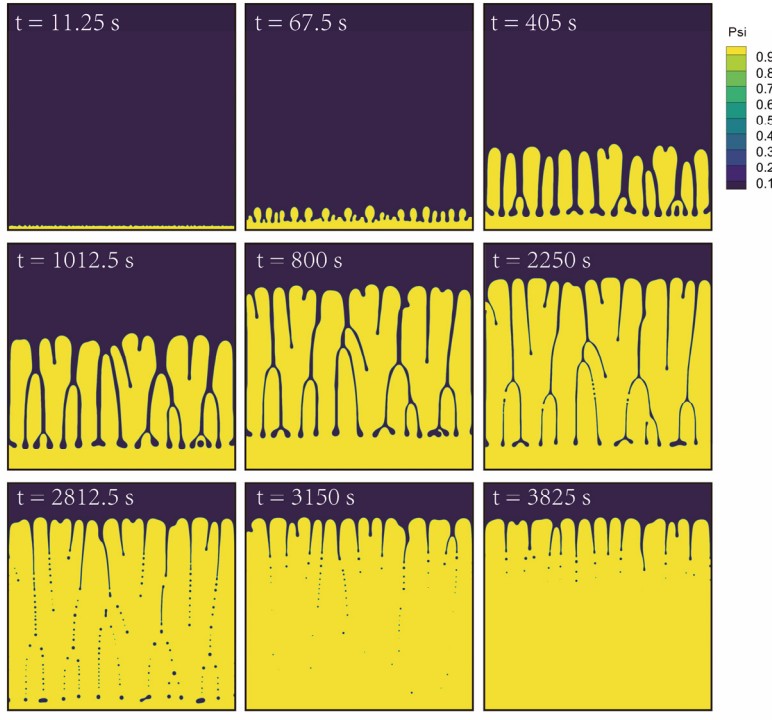

**Figure 3.** Microstructure growth simulation of bismuth telluride-based crystal material.

At the current cooling rate, the bismuth telluride material presents a cellular crystal growth mode. It should be noted that in addition to cellular crystal growth, the planar interface growth and dendritic crystal growth are possible growth modes according to crystal growth theory [36,37]. The growth morphology of crystal structure mainly depends on the temperature gradient and growth rate.

## 4. Results and Discussion

### 4.1. Simulation of Heat Flow Distribution and Microstructure Growth in Zone Melting Process

The temperature distribution and exchange in the zone melting process of bismuth telluride-based crystal determine whether the solidification is stable or not and is a deciding factor in obtaining high-quality thermoelectric semiconductor materials. Figure 4a shows the position of the melting furnace, and Figure 4b,c shows the external and internal temperature distribution of the material, respectively. The surface temperature distribution of the material is uniform in the horizontal direction. However, inside the material, the temperature gradient is not uniform but varies with the distance from the center of the heat source. Near the center of the heat source, the isotherm surface is convex toward the heat source, while away from the center of the heat source, the isothermal surface is concave. The morphology of the isothermal surface is roughly symmetrical along the center of the heat source.

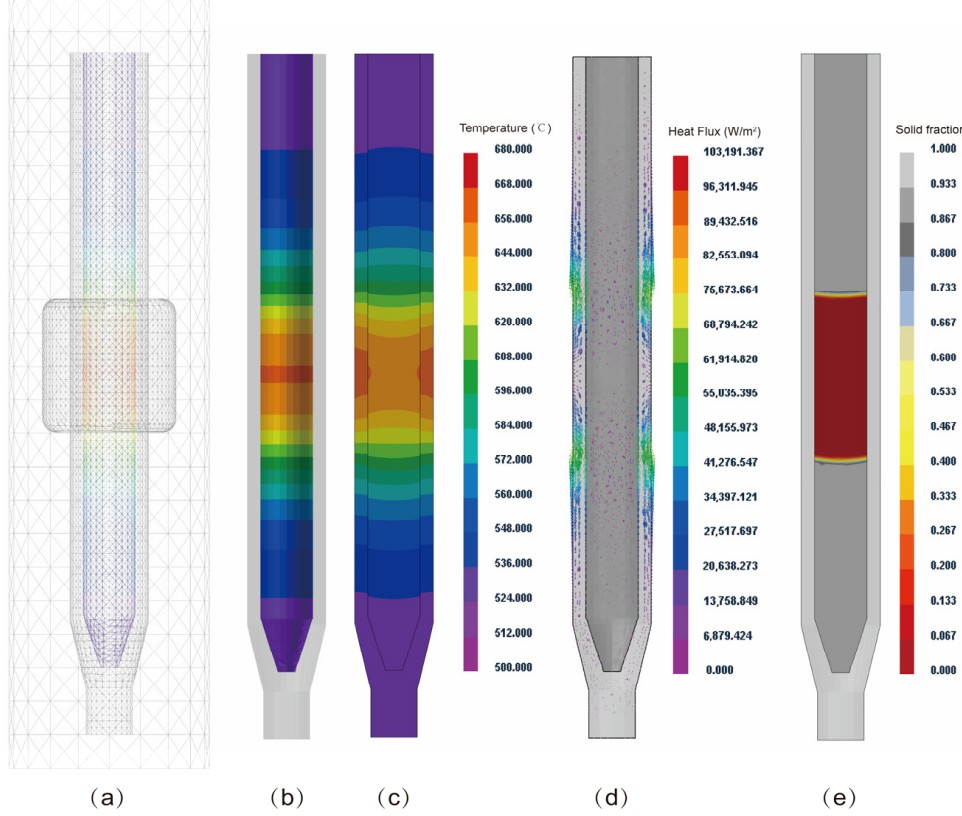

**Figure 4.** Heat flow and temperature analysis during zone melting process of bismuth telluride-based crystal material. (**a**) Simulation grid, (**b**) surface temperature distribution, (**c**) cross section temperature distribution, (**d**) heat flow analysis, (**e**) melting zone analysis.

This morphology change of isothermal surface is caused by the movement of heat flow. In addition to conduction to both ends, a considerable part of the heat absorbed by the ampoule dissipates to the environment through thermal radiation or convection. Figure 4b shows the heat flow distribution of the zone melting process, where the strongest heat flow loss of the ampoules is on both sides of the melting furnace. Thus, the ampoule contains both the highest density of energy absorption and the highest density of heat loss near

the heating zone of the melting furnace. Because of the large heat exchange around the interface of the heating area, when the temperature conditions are not controlled properly, the temperature field is prone to fluctuate, resulting in the instability of the interface. At the area opposite the heating furnace, the heat flow direction of the ampoule is from the outside to the inside. In this area, the surface temperature of the material is higher than the inside, thus causing the isothermal surface to bulge toward the center of the heat source. On both sides of the heating area, heat dissipates intensely, leading to the surface temperature of the ampoule lower than the inside and finally causing the isothermal surface to be concave in the center of the heat source. The morphology of temperature fields will eventually affect the solidification interface shown in Figure 4e. A smoother solidification interface is expected in the zone melting process of crystals to prevent crack and other defects, and hence, it is very necessary to continuously adjust and optimize the zone melting process to reduce the disturbance of heat flow around the interface.

Figure 5 shows the morphology of the solidification interface. The length of the region with a solid fraction between 0 and 1, which is called mushy zone, is about 1.5 mm. Six temperature observation points are arranged sequentially in the mushy zone according to the distance from the center position. Figure 5c shows the temperature changes of these six observation points during the zone melting process. There is no significant difference in their maximum temperature but a slight difference in time. This indicates that in this zone melting process, the maximum temperature that can be reached in the areas with different distances from the surface to the center is similar, while the time to reach the maximum temperature is different. Areas close to the material surface reach maximum temperature earlier.

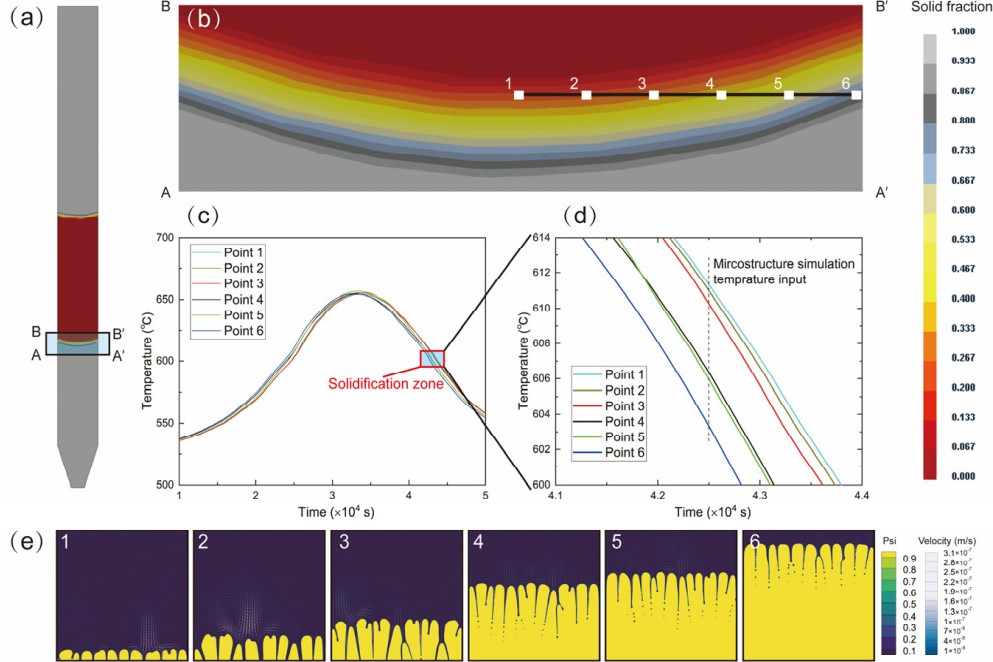

**Figure 5.** Macro-micro-coupling simulation of zone melting process of bismuth telluride-based crystal materials. (**a**) Macro temperature field simulation of the zone melting process, (**b**) the morphology of solidification interface and the position of six temperature observation points, (**c**) the temperature change profile of observation points throughout the zone melting process, (**d**) the temperature change profile near the melting point, (**e**) microstructure growth simulation.

Figure 5d shows the variation of the temperature at the observation points near the melting point. The temperature variation is taken as the temperature boundary condition for the microstructure simulation. The microstructure simulation result is shown in Figure 5e. The white arrows in Figure 5e refer to the flow velocity of the liquid phase.

The liquid phase flows more intensely near the cellular crystal interface, which accelerates the distribution of solutes in the liquid phase and thus further accelerates the solidification.

### 4.2. Influence of Zone Melting Process on Melt Zone Characteristics
#### 4.2.1. Influence of Melting Temperature on Melt Zone Characteristics

The zone melting process determines the distribution of the temperature field and thus determines whether the solidification interface is continuously stable. An unstable solidification interface leads easily to cracks and other defects and eventually results in the degradation of material properties or even scrap. Figure 6a shows the morphologies of the melting zone at different melting temperatures. The moving velocity of the melting furnace is 5 mm/h. The melting zone morphologies intuitively reveal the fact that the higher the melting temperature, the longer the length of the melting area. Figure 6b shows the curvatures of the melting interface and solidification interface of the melting zones. The curvature results show that the melting temperature mainly affects the melting interface and has a relatively lower impact on the solidification interface. The melting zone under higher melting temperature has a smaller interface curvature of the melting zone and thus a smoother interface. While on the contrary, this phenomenon is just the opposite for the solidification interface. The melting zone under higher melting temperature has a higher interface curvature of the melting zone. Compared with the melting interface, the solidification interface has a more important effect on acquiring defect-free and high-performance bismuth telluride crystal materials. In the case that the melting temperature can sufficiently melt the bismuth telluride material, the melting zone with a lower melting temperature tends to obtain a smooth solidification interface.

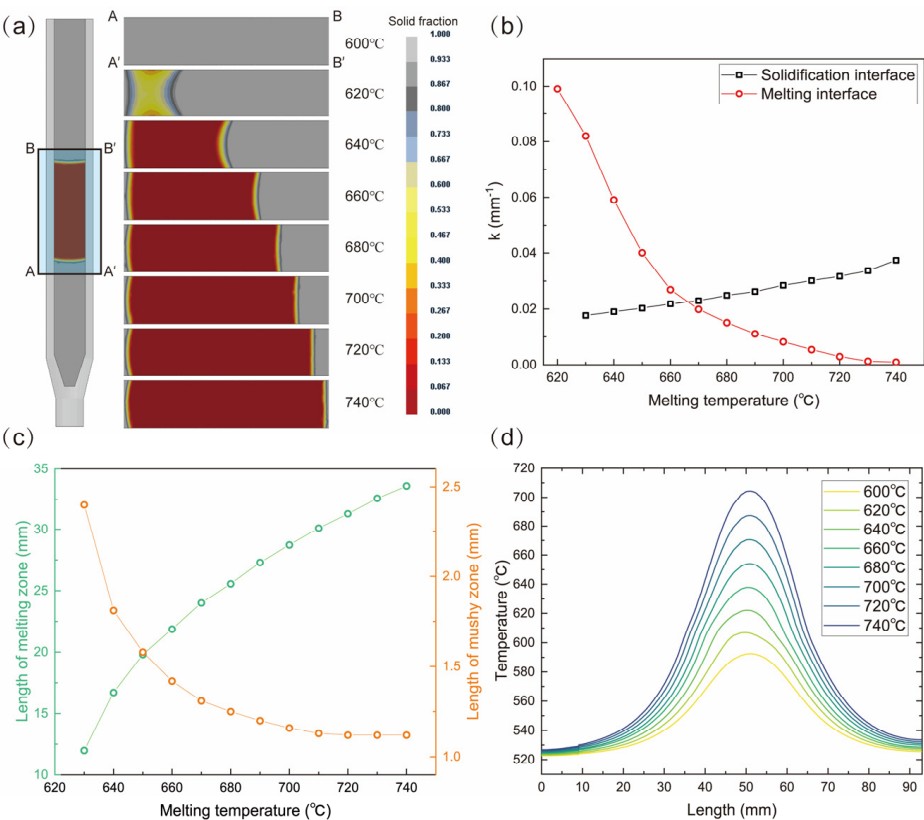

**Figure 6.** Influence of melting temperature on melting zone characteristics and temperature distribution. (**a**) Morphology of melting zone under different melting temperatures, (**b**) curvature of solidification interface and melting interface, (**c**) length of melt zone and mushy zone, (**d**) temperature distribution at the center axis.

This phenomenon can be explained by the heat flow distribution results and latent heat theory in Figure 6d. The melting furnace continuously supplies the heat required for melting. The temperature of the area covered by the heat source in the ampoule can be guaranteed to melt the bismuth telluride. Since the quartz tube dissipates heat significantly to the environment on both sides of the heating zone, the heat inside the bismuth telluride will be rapidly transferred outwards on both sides, resulting in heat loss, thus forming an inwardly concave melting interface. Due to the latent heat of solidification, the bismuth telluride-based crystal material needs to absorb more heat to melt the material, so this phenomenon is aggravated on the melting side. Moreover, when under lower melting temperatures, the influence of latent heat on the interface will be more obvious. On the solidification interface side, due to the influence of latent heat of solidification, the material will release heat during solidification and keep the local temperature stable, which weakens this phenomenon. Therefore, the solidification interface is more stable than the melting interface when under relatively low melting temperature.

When the temperature of the melting temperature increases, the latent heat of the melting interface side is replenished in time, so the interface is gradually leveled. On the solidification interface side, as the melting zone becomes longer and more bismuth telluride-based crystal material is melted, the heat released from the latent heat of solidification is even greater than the heat lost in some parts, which leads to the gradual increase of the solidification curvature. The length of the melting zone and the mushy zone are shown in Figure 6c. With the increase in temperature, the heat supply becomes sufficient. The length of the melting zone continues to increase. While the length of the mushy zone decreases with the increase of the heat source temperature, which is related to the temperature gradient inside the bismuth telluride-based crystal material. When under higher temperatures, the melting zone possesses a larger internal temperature gradient, resulting in a shorter mushy zone. The temperature distribution at the central axis of the material at different melting temperatures is shown in Figure 6d. Under different melting temperatures, the temperature distribution profiles are similar, but the maximum temperature that the material can reach is obviously different.

### 4.2.2. Influence of Melting Furnace Moving Velocity on Melt Zone Characteristics

For the large-scale production of crystal bismuth telluride, faster melting furnace moving velocity can speed up the production and reduce the energy loss. However, when the moving velocity is too fast, the solidification interface will be unstable, and the phenomenon of component supercooling may occur, resulting in the generation of the crack defect. Therefore, to reduce economic loss, it is very necessary to optimize the moving velocity of the melting furnace before production. Figure 7a shows the simulation morphologies of the melting zone under different moving velocities of the melting furnace. The heat source temperature is 680 °C in these simulations. The length of the melt zone and the curvature of the interface are shown in Figure 7b,c, respectively. Contrary to the heat source temperature, the moving velocity of the heat source tends to affect the curvature of the interface but has little effect on the length of the melting zone. The curvature of both the solidification interface and the melting interface increase with the melting furnace moving velocity. With the melting furnace moving velocity less than 10 mm/h, the length of the solidification interface and mushy zone can be guaranteed at a small value.

Figure 7d shows the temperature distribution of the central axis of the material under different moving velocities of the melting furnace. There are two small plateaus in the temperature curve as the moving velocity of the heat source is higher than 20 mm/h. The plateau becomes more obvious with the increase of heat source moving velocity. These plateaus correspond to their melting and solidification temperatures. The generation of the plateaus, we believe, is related to latent heat. The excessive moving speed makes the melting and solidification interface unstable. With the unstable interface, the latent heat effect is further amplified and eventually leading to the formation of two obvious temperature plateaus.

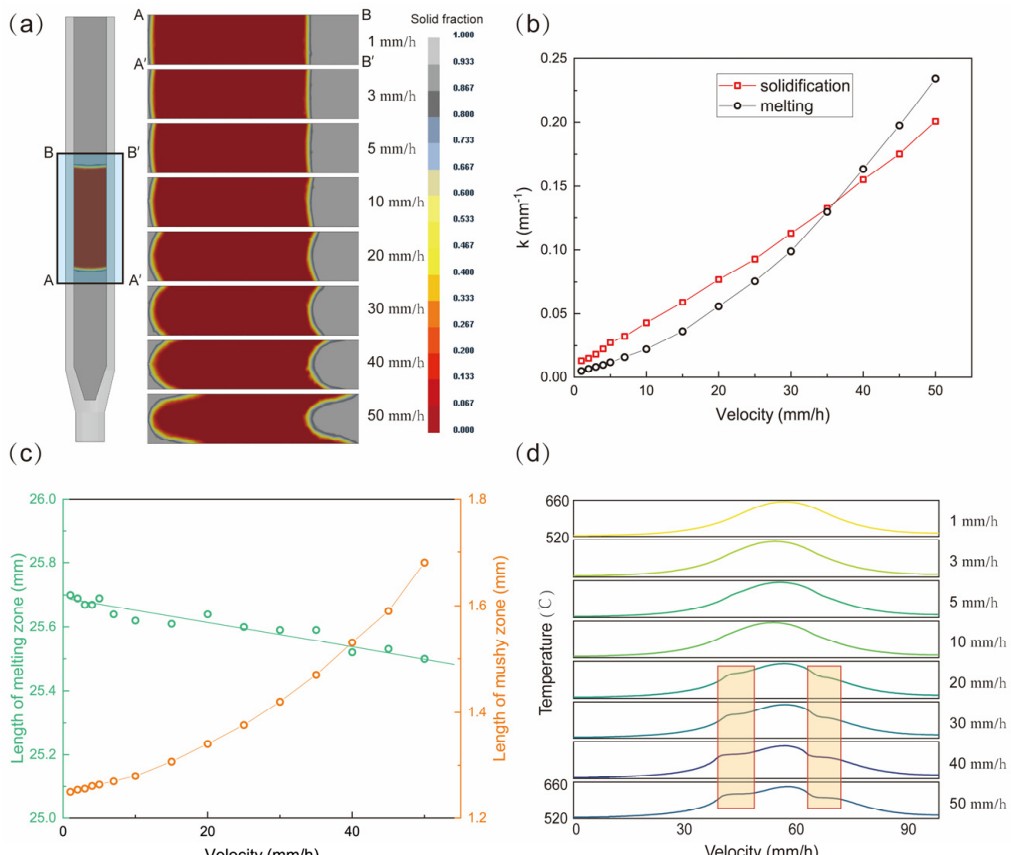

**Figure 7.** Influence of melting furnace moving velocity on melting zone characteristics and temperature distribution. (**a**) Morphology of melting zone under different melting temperatures, (**b**) curvature of solidification interface and melting interface, (**c**) length of melt zone and mushy zone, (**d**) temperature distribution at the center axis.

The microstructure growth of a bismuth telluride crystal at different heat source moving velocities was further simulated, as shown in Figure 8. The simulation results are consistent with the crystal growth theory [36,37] and experimental results [38]. Under the condition of the constant temperature gradient, with the increase of growth rate, the crystal grows in the form of a flat interface, then, transitions into the form of a cellular crystal, and, finally, into the form of dendrite growth. This transformation is mainly affected by temperature and composition. Figure 9 shows the dendrite spacing and height of the bismuth telluride cellular crystals at different moving velocities. With the moving velocity of the melting furnace less than 12 mm/h, the dendrite spacing of the cellular crystals does not change much, while the height of the dendrites changes significantly. Due to the existence of component supercooling, there is a certain difference in the composition of the first-solidified material and the later-solidified material on the same solidification plane, which results in the micro-segregation phenomenon inside the material. However, the excessively large dendrite spacing and dendrite height will aggravate the micro-segregation and lead to the uneven distribution of elements. Therefore, in order to obtain bismuth telluride thermoelectric materials with uniform composition distribution and high performance, it is necessary to control the growth morphology of bismuth telluride crystals to grow in the form of the flat interface or in the form of the cellular crystal with a small dendrite height.

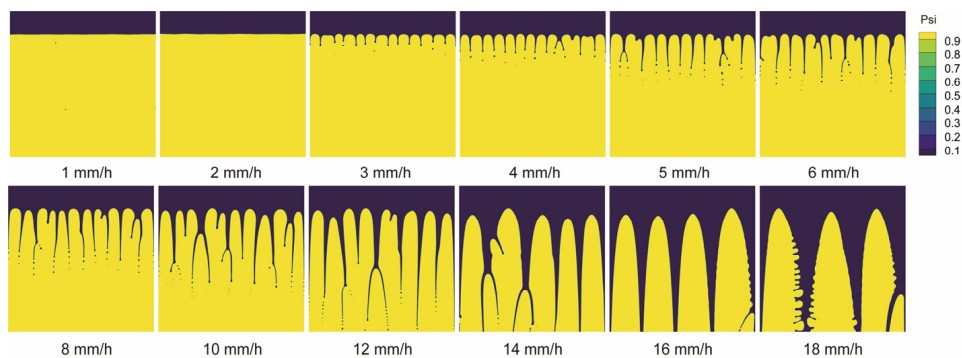

**Figure 8.** Microstructure growth simulation of bismuth telluride crystals at different melting furnace moving velocities.

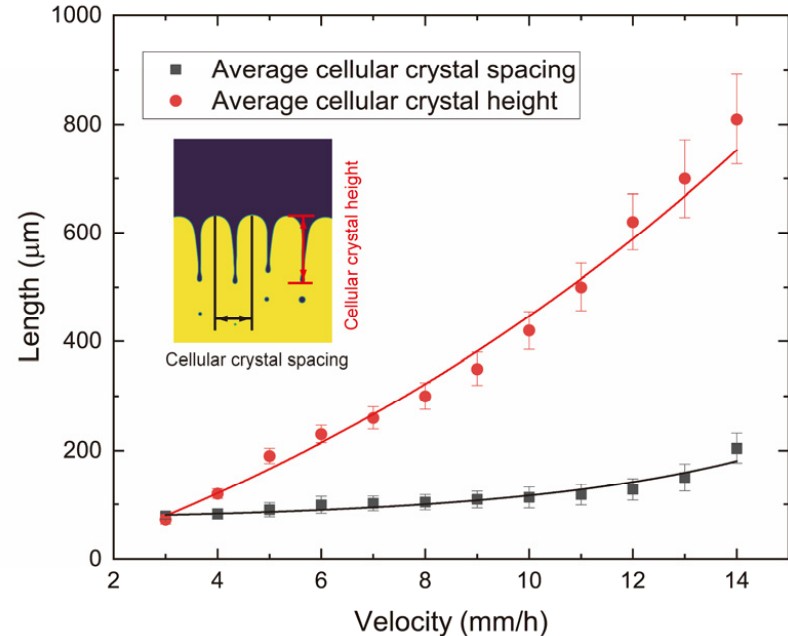

**Figure 9.** Effect of melting furnace moving velocity on dendrite spacing and height of bismuth telluride cellular crystal.

### 4.2.3. Zone Melting of Bismuth Telluride-Based Crystal Materials under Space Microgravity

In order to investigate the influence of gravity on the zone melting process of bismuth telluride-based crystal material, the zone melt experiments were carried out both under microgravity condition on the Tiangong II space laboratory and conventional gravity condition on the ground. According to the simulation results, the appropriate melting temperature and melting furnace moving velocity are chosen, which are 660 °C and 5 mm/h, respectively.

The element distribution in the zone melting samples under space microgravity and ground conventional gravity are shown in Figure 10. Table 2 shows the standard deviation of element distribution of the materials fabricated in space and on ground. The standard deviation of the element distribution of the material fabricated on the ground is greater than that in the space, indicating the element distribution of the material prepared on the ground is more inhomogeneous. While under the microgravity condition of the space station, the convection of components is significantly suppressed, and the distribution of elements is more uniform. Therefore, the chemical uniformity of the materials fabricated in the space is better than that on the ground.

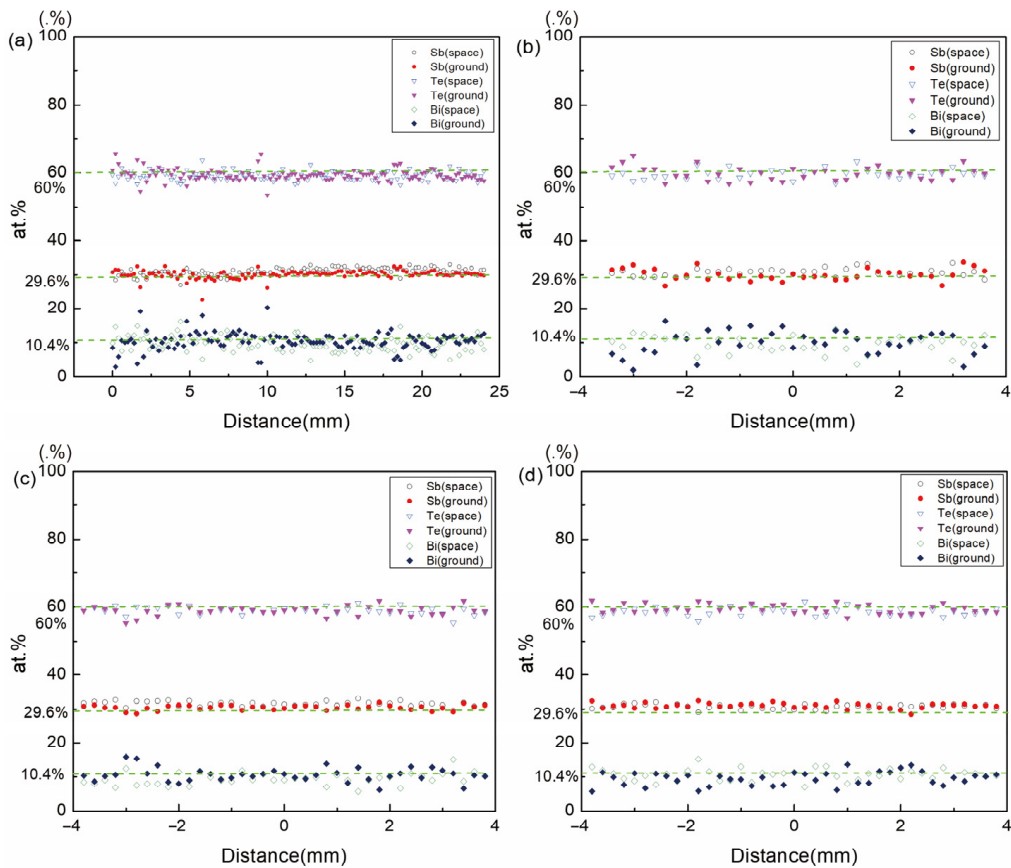

**Figure 10.** Atomic composition profile of crystals (**a**) along the growth direction, along the radial direction (**b**) at L = 5 mm, (**c**) at L = 15 mm, (**d**) at L = 24 mm. L represents the distance between the observation position and the starting position of zone melting.

**Table 2.** Standard deviation of element distribution in bismuth telluride-based crystal materials prepared in space and on ground.

|  | **Bi(Space)** | **Bi(Ground)** | **Sb(Space)** | **Sb(Ground)** | **Te(Space)** | **Te(Ground)** |
|---|---|---|---|---|---|---|
| Standard deviation | 2.16 | 2.54 | 1.08 | 1.20 | 1.32 | 1.68 |

The surface morphologies of the bismuth telluride-based crystal material fabricated on the ground and in space are shown in Figure 11a1,a2, respectively. It can be seen intuitively from the figure that the surface quality of the material prepared in the space is relatively poor, which is also confirmed by the CT results of Figure 11b1,b2. During the crystal growing experiment, the ingots become liquid, and some liquid moves past the quartz cotton into the fixating quartz tube due to wetting phenomena and capillarity. For the space experiment, the liquid moves into the fixating quartz tube due to the capillarity effect. Large cavities remain after the crystal growing experiment. That is the reason why the space-grown crystal has large cavities. While for the ground experiment, less liquid moved into fixating quartz tube in comparison with the space experiment because of suppression of capillarity by gravity. So, the ground-grown crystal does not have large cavities.

Figure 11c1,c2 shows the results of the Seebeck coefficient distribution. The scanning intervals of the Seebeck coefficient distribution result are 1 mm and 0.5 mm along the growth direction and radial direction, respectively. The test temperature is 25 °C. On the whole, the Seebeck coefficient of ground samples is higher than that of space samples, but the possibility that the low Seebeck coefficient of the space sample is caused by surface quality cannot be ruled out. In the early stage of growth, the Seebeck coefficient of ground samples fluctuated greatly. The distribution of the Seebeck coefficient of the space sample is more

uniform than that of the ground sample, mainly due to the suppression of the convection component under the condition of microgravity and the higher chemical uniformity. In conclusion, the surface quality of the sample prepared in the space is relatively poor, but the chemical uniformity and performance uniformity are better than that on the ground.

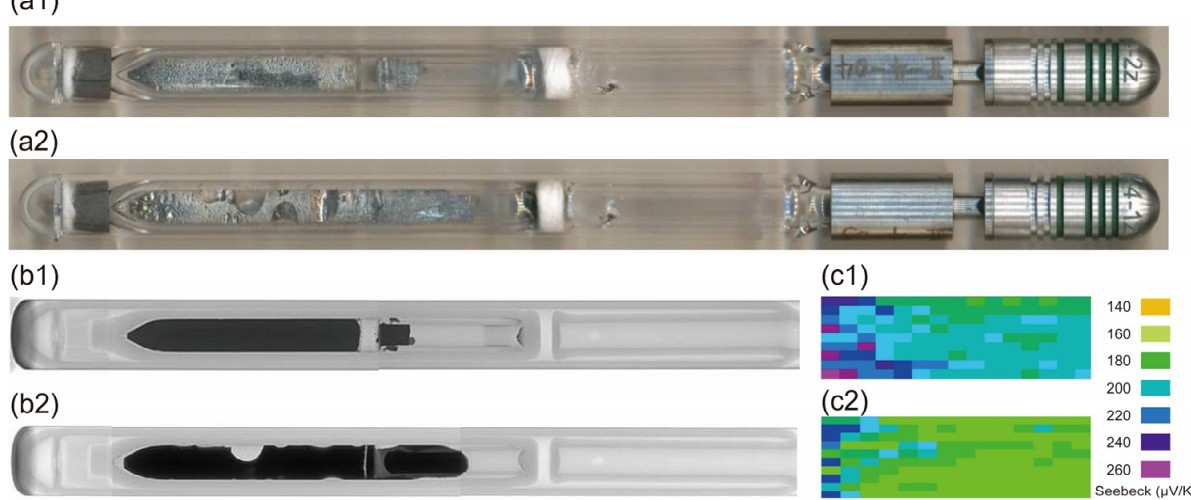

**Figure 11.** Comparison between bismuth telluride-based crystal materials prepared on ground and in space. (**a1**) Morphology of the sample prepared on the ground, (**a2**) Morphology of the sample prepared in the space, (**b1**) CT result of the sample prepared on the ground, (**b2**) CT result of the sample prepared in the space, (**c1**) Seebeck coefficient distribution of the sample prepared on the ground, (**c2**) Seebeck coefficient distribution of the sample prepared in the space.

## 5. Conclusions

This work studied the zone melting process of bismuth telluride-based crystal material by numerical simulation and experimental methods. The main conclusions are as follows:

1.  A macro-micro-coupled simulation model was established to analyze the distribution of temperature and heat flow during the zone melting process. The influence of latent heat on solidification interface morphology is discussed.
2.  The influence of the zone melting process on melting zone characteristics has been simulated. The melting temperature tends to affect the length of the melting zone, while the moving velocity of the melting furnace tends to affect the curvature of the melting and solidification interface.
3.  From the perspective of latent heat, the curvature difference between solidification interface and melting interface at different melting temperatures and the plateau phenomenon on the temperature curve of the central axis of bismuth telluride ingot at different heat source moving speeds are explained.
4.  In order to investigate the influence of gravity on the zone melting process of bismuth telluride-based crystal material, zone melt experiments were carried out both under microgravity condition on the Tiangong II space laboratory and conventional gravity condition on the ground. The comparison results show that the surface quality of the sample prepared in the space is relatively poor, but the chemical uniformity and performance uniformity are better than that on the ground.

**Author Contributions:** Simulation, H.X.; experiment, X.L.; writing—original draft preparation, H.X.; writing—review and editing, Q.X. and X.L. All authors have read and agreed to the published version of the manuscript.

**Funding:** This research was supported by the China Space Station Project (TGJZ80701-2-RW024), the Strategic Priority Research Program on Space Science, the Chinese Academy of Sciences (XDA15051200), and the National Natural Science Foundation of China (U1738114).

**Institutional Review Board Statement:** Not applicable.

**Informed Consent Statement:** Not applicable.

**Data Availability Statement:** Not applicable.

**Conflicts of Interest:** The authors declare no conflict of interest.

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
