# Peer review of "Macro-Micro-Coupling Simulation and Space Experiment Study on Zone Melting Process of Bismuth Telluride-Based Crystal Materials"

_metals, doi:10.3390/met12050886_

Round 1

Reviewer 1 Report

Very well written article which includes simulation with experimental verification. I suggest to improve figures quality. A slight correction of the text is needed in terms of editing (eg Kelvin as "k"). 

Author Response

Point 1: Very well written article which includes simulation with experimental verification. I suggest to improve figures quality. A slight correction of the text is needed in terms of editing (eg Kelvin as "k").

Response 1:  Dear reviewer, thank you for your kind suggestion. The figures in the manuscript have been replaced with high-quality figures, with at least 1000 ppi. The content has also been further modified in detail.

Reviewer 2 Report

The review is attached as a pdf file.

Author Response

Point 1: The abstract is too general. It only roughly describes what has been measured for refined composite materials. The abstract should contain the most important conclusions from the conducted research. It is worth including the values of some parameters in the abstract. Please rewrite the abstract so that it meets the basic requirements.

Response 1:  Dear reviewer, thank you for your kind suggestion. The abstract is completely rewritten. In addition to describing the work of this paper, the relevant conclusions are also written in the abstract.

Point 2: There are many editorial errors in the paper e.g. no subscript, inconsistent use of lowercase/uppercase and italics. Please correct these.

Response 2:  The full manuscript was carefully checked. The editing errors modified are as follows:

Table 1:

W/(m·k)  ->  W/(m·K)

g/m3  ->  g/m3

m4/ (j·s)  ->  m4/ (j·s)

4.0 x 10-11  ->  4.0 x 10-11

Introduction:

Konig  ->  König

softwares  ->  software

have  ->  has

Simulation:

literature  ->  literatures 

Procast  ->  ProCAST

Point 3: The quality of all the figures posted is very low. Practically nothing can be read from them. Please improve it.

Response 3:  The figures in the new manuscript have been replaced with high-quality figures (at least 1000 ppi).

Point 4: Introduction

Response 4:  The introduction is checked, and some errors have been revised.

Point 5: Page 3, Experiment: There is a lack of basic information about each step of the research that is necessary to verify that such measurements can be made:

  1. a) What materials were used to synthesize the samples- manufacturer, purity of samples?
  2. b) Was the zone melting furnace made in-house by the authors or did they use a commercial device?
  3. c) How was the zone melting furnace at the space station constructed?
  4. d) Were the zone melting furnace used on the space station and on earth the same?
  5. e) What was the shape and dimensions of the sample on which the measurements were made?
  6. f) How does the device work YXLON MU2000-D225 CT?

Response 5:  Thank you for your kind reminding. The information was added in the Experiment part:

  1. High-purity of Bi (5N) , Sb (5N) and Te (5N) granules were weighed according to the stoichiometric ratio of Bi52Sb1.48Te3.
  2. The space experimental furnace is custom-developed by Shanghai Institute of Silicate.
  3. There are heating units, pulling device and temperature detection system in the experimental furnace to maintain the directional temperature gradient.
  4. The zone melting experiments were carried out under conventional gravity condition on the ground and microgravity condition on the Tiangong II space laboratory with same experimental furnaces.
  5. The length of the solidified sample is 55mm, and the diameter is 7.8mm…. The detection area is the axial section of ingot. The starting position of measurement is 5mm from the bottom of the ingot.
  6. The samples were characterized by a YXLON MU2000-D225 CT analysis system. The X-ray produced by CT analysis system can penetrate ampoules and bismuth telluride materials, and provide interior morphology of the ingots.

Point 6: X-ray analysis of the materials obtained is missing in this paper. I believe that such analysis is mandatory in this type of research, because even small amounts of foreign phases significantly affect the thermoelectric properties of the compound.

Response 6:  Dear reviewer, this research may be different from the component research of bismuth telluride materials. In the component research of bismuth telluride materials, there is a lot of work focusing on the thermoelectric performance of bismuth telluride materials with different components. Their research use XRD analysis to characterize crystal structure in different components. Such as :

Figure 1 (a) Powder XRD patterns of Bi0.5Sb1.5Te2.96+x (x = 0–0.12) compounds

(Huang H, Li J, Chen S, et al. Anisotropic thermoelectric transport properties of Bi0. 5Sb1. 5Te2. 96+ x zone melted ingots[J]. Journal of Solid State Chemistry, 2020, 288: 121433.)

Figure 2 XRD patterns of various compositions of BixSb2-xTe3 synthesized by thermal explosion.

  • 2θ in the range of 10°~80°; (b) 2θ in the range of 57°~65°.

(Zheng G, Su X, Xie H, et al. High thermoelectric performance of p-BiSbTe compounds prepared by ultra-fast thermally induced reaction[J]. Energy & Environmental Science, 2017, 10(12): 2638-2652.)

However, if the literature does not involve any components research, the XRD not be necessary. Such as:

  • Yamashita O, Sugihara S. High-performance bismuth-telluride compounds with highly stable thermoelectric figure of merit[J]. Journal of Materials Science, 2005, 40(24): 6439-6444.
  • Biswas K G, Sands T D, Cola B A, et al. Thermal conductivity of bismuth telluride nanowire array-epoxy composite[J]. Applied Physics Letters, 2009, 94(22): 223116.
  • Chen Y R, Hwang W S, Hsieh H L, et al. Thermal and microstructure simulation of thermoelectric material Bi2Te3 grown by zone-melting technique[J]. Journal of crystal growth, 2014, 402: 273-284.
  • Zheng G, Su X, Li X, et al. Toward high‐thermoelectric‐performance large‐size nanostructured BiSbTe alloys via optimization of sintering‐temperature distribution[J]. Advanced energy materials, 2016, 6(13): 1600595.
  • Carter M J, El-Desouky A, Andre M A, et al. Pulsed laser melting of bismuth telluride thermoelectric materials[J]. Journal of Manufacturing Processes, 2019, 43: 35-46.
  • Yin Z, Zhang X, Wang W, et al. Melt Growth of Semiconductor Crystals Under Microgravity[M]//Physical Science Under Microgravity: Experiments on Board the SJ-10 Recoverable Satellite. Springer, Singapore, 2019: 327-360.
  • Wang W, Li X, Gu M, et al. Low Temperature Joining and High Temperature Application of Segmented Half Heusler/Skutterudite Thermoelectric Joints[J]. Materials, 2019, 13(1): 155.
  • Qiu J, Yan Y, Luo T, et al. 3D Printing of highly textured bulk thermoelectric materials: mechanically robust BiSbTe alloys with superior performance[J]. Energy & Environmental Science, 2019, 12(10): 3106-3117.

In this research, the components are completely the same in the experiments both on the ground and in space, which is Bi0.52Sb1.48Te3. The powder XRD results will be exactly the same. This research focuses more on the influence of composition distribution and does not study bismuth telluride materials with different components. Therefore, we think the XRD analysis may not be necessary in this study.

Point 7: The conducted research concerns Bi2Te3, which is classified as a semiconductor (to be more precise we should say that it is a topological insulator), therefore I do not understand why these results are proposed for publication in the journal "Metals". The journal "Metals", as the name suggests, is supposed to deal with metallic materials. It is a bit like publishing articles about cats in a journal about dogs. The readers will not like it.

Response 7:  Thank you for your doubt. We also thought deeply about this question, but After a long time of thinking and hesitating, we finally chose the "Metals" as the journal to submit.

In this research, we use ProCAST and phase-field simulation model to study the solidification of Bi0.52Sb1.48Te3.

The ProCAST is commercial software which widely applied in metal casting field. In other research fields, we think there will be few people may know this software due to its specialization. According to our investigation, because of specific optimization work of ProCAST, it is much more suitable for the simulation of zone melting process, campared with other general fluent-solidification simulation software in non-casting field such as Fluent, Fluent3D, CFX, Comsol.

Crystal phase-field simulation is also well received research method widely applied in the field of metal solidification, which can simulate the microstructure evolution. However, according to our investigation results, there are few specific phase-field simulation models or literatures in the field of zone smelting of bismuth telluride materials. In bismuth telluride materials field, phase-field simulation may not be a well known method.

This manuscript focuses more on the research method of numerical simulation and the corresponding mathematical model. We want to express the fact that due to the similar solidification process and crystal structure of the material, zone melting process of bismuth telluride materials can also be studied by the simulation method which is commonly used in the metal field.

Therefore, we think it may be more appropriate to submit this manuscript on "Metals".

Reviewer 3 Report

Zone melting is one of the main ways to obtain materials. The method of numerical simulation of directional solidification allows you to simulate temperature and microstructure changes during solidification, which accelerates research progress in this area. Thermoelectric materials based on bismuth telluride are currently in demand. All this speaks of relevance. This article presents a macro-micro coupled simulation model for modeling the process of zone melting of crystalline materials based on bismuth telluride, which allows to reduce material costs. Heat flow distribution, solidification surface curvature and microstructure growth during zone melting were modeled to study the effect of melting temperature and melting furnace speed on solidification. The paper provides a good literature review and describes in detail the simulation of zone melting with a large number of diagrams, diagrams and drawings. All drawings are informative. It is interesting to study the effect of gravity on the process of zone melting of a crystalline material based on bismuth telluride. The only remark is that it is interesting to compare simulation and experimental data. But this does not detract from the dignity of the article. The work left a very good impression. It is understandable to a researcher who is just starting work in this direction. I recommend this work for publication.

Author Response

Point 1: Zone melting is one of the main ways to obtain materials. The method of numerical simulation of directional solidification allows you to simulate temperature and microstructure changes during solidification, which accelerates research progress in this area. Thermoelectric materials based on bismuth telluride are currently in demand. All this speaks of relevance. This article presents a macro-micro coupled simulation model for modeling the process of zone melting of crystalline materials based on bismuth telluride, which allows to reduce material costs. Heat flow distribution, solidification surface curvature and microstructure growth during zone melting were modeled to study the effect of melting temperature and melting furnace speed on solidification. The paper provides a good literature review and describes in detail the simulation of zone melting with a large number of diagrams, diagrams and drawings. All drawings are informative. It is interesting to study the effect of gravity on the process of zone melting of a crystalline material based on bismuth telluride. The only remark is that it is interesting to compare simulation and experimental data. But this does not detract from the dignity of the article. The work left a very good impression. It is understandable to a researcher who is just starting work in this direction. I recommend this work for publication.

Response 1:  

Dear reviewer, thank you for your kind recognition and encouragement.

We are in a lab which mainly studies in numerical simulation technology of solidification process. We do lack experience in bismuth telluride materials. This manuscript is our first attempt to the research of the zone melting process of bismuth telluride material. In the attempt, we did encountered many problems and tried our best to fix these problems. In the future research, we will further optimize our model. We hope we can bring a little help to the field of bismuth telluride solidification process.

Again, thank you for your support, it means a lot to us.

Round 2

Reviewer 2 Report

The review is attached as a pdf file.

Author Response

Response to Reviewer 2 Comments

Point 1: Abstract, line 12: „In this research, A macro-micro” - > In this research, a macro-micro

Response 1: Dear reviewer, thank you for pointing out this editorial mistake. The mistake is changed in the new manuscript.

Point 2: Page 1, line 38: The symbols e and l should be written as subscript.

Response 2: Dear reviewer, symbols are changed in the new manuscript.

Point 3: In many cases, the space between the test and the square brackets with the reference number is missing (e.g., page 1 line 42).

Response 3: Dear reviewer, all missing spaces have been added.

Point 4: For many figure captions, the dots at the end are missing.

Response 4: The dots are added in the captions of all figures and tables.

Point 5: Page 5. Line 177. A dot is missing after „as shown in Figure 2”

Response 5: Dear reviewer, the dot is added in after the sentence. Thank you.

Point 6: Page 5. Line 180. There is no panel (c) on Figure 2.

Response 6: Thank you for pointing out this critical error. Figure 2 (b) and Figure 2 (c) are changed to Figure 2 (a) and Figure 2 (b).

Point 7: References: For many compound names, the lower indices are missing e.g. [4].

Response 7: Dear reviewer, all the references have been modified.

[4] Bi2Te3

[9] BixSb2-xTe3

[12] Bi2Te3 and Sb2Te3

[13] Bi2Te3/Sb2Te3

[15](Bi2Te3)90 (Sb2Te3)5 (Sb2Se3)5 and(Sb2Te3)72 (Bi2Te3)25 (Sb2Se3)3

[16] Bi0.5Sb1.5Te3

[17] (Bi2Te3) x(Sb2Te3)1-x

[18] Bi0.5Sb1.5Te3

[23] Bi2Te3

[34] Bi2Te3-Bi2Se 3-Bi2S3

[38] Bi2-xSbxTe3

Thank you for your careful and kind review, it helps a lot to us.